# Persistent, Asymptomatic Colonization with *Candida* is Associated with Elevated Frequencies of Highly Activated Cervical Th17-Like Cells and Related Cytokines in the Reproductive Tract of South African Adolescents

Anna-Ursula Happel,[a] Melanie Gasper,[b] Christina Balle,[a] Iyaloo Konstantinus,[a,c] Hoyam Gamieldien,[a] Smritee Dabee,[b] Katherine Gill,[d] Linda-Gail Bekker,[d] Jo-Ann S. Passmore,[a,e,f] Heather B. Jaspan[a,b,g]

[a]Department of Pathology, Institute of Infectious Disease and Molecular Medicine, University of Cape Town, Cape Town, South Africa
[b]Seattle Children's Research Institute, Seattle, Washington, USA
[c]Namibia Institute of Pathology, Windhoek, Namibia
[d]Desmond Tutu HIV Centre, University of Cape Town, Cape Town, South Africa
[e]DST-NRF CAPRISA Centre of Excellence in HIV Prevention, Cape Town, South Africa
[f]National Health Laboratory Service, Cape Town, South Africa
[g]Department of Pediatrics and Global Health, University of Washington, Seattle, Washington, USA

**ABSTRACT** Cervicovaginal inflammation, nonoptimal microbiota, T-cell activation, and hormonal contraceptives may increase HIV risk, yet associations between these factors and subclinical *Candida* colonization or hyphae are unknown. We collected cervicovaginal samples from 94 South African adolescents, aged 15 to 19 years, who were randomized to injectable norethisterone enanthate (Net-En), an etonorgesterol/ethinyl estradiol vaginal ring (NuvaRing), or oral contraceptives in the UChoose trial (NCT02404038) at baseline and 16 weeks post-randomization. We assessed cervicovaginal samples for subclinical *Candida* colonization (by quantitative PCR [qPCR]), hyphae (by Gram stain), microbiota composition (by 16S rRNA gene sequencing), cytokine concentrations (by Luminex), and cervical T-cell phenotypes and activation (by multiparameter flow cytometry). While hormonal contraceptive type did not influence incidence of *Candida* colonization or hyphae, hyphae presence was associated with significantly elevated concentrations of IL-22, IL-17A and IL-17F, all produced by Th17 cells, but not of other cytokines, such as IL-1$\beta$ or IL-6, after adjustment for confounders. Subclinical *Candida* colonization was associated with reduced frequencies of Th17-like cells and elevated frequencies of CCR6-CCR10 T cells. Women with *Candida* hyphae were less likely to have bacterial vaginosis (BV). Persistent, subclinical colonization with *Candida* over 16 weeks was associated with significant increases in Th17-related cytokine concentrations and highly activated Th17-like and CCR6-CCR10 T-cell frequencies. These data suggest that vaginal *Candida* colonization and hyphae increase Th17-related cytokines, but not overall female genital tract inflammation in Sub-Saharan African adolescents. Persistent *Candida* colonization, even when asymptomatic, may increase Th17 cell frequencies and related cytokines and thereby could subsequently increase HIV risk, although the causal relationship requires confirmation.

**IMPORTANCE** Sub-Saharan African female adolescents are globally at the highest risk of HIV acquisition, and genital inflammation, microbial dysbiosis, and cervical HIV target cell activation are thought to contribute to this risk. Previously, the relationship between these mucosal factors and subclinical vaginal *Candida* colonization or hyphae has not been described, and the role of HIV-susceptible Th17 cells in mediating anti-*Candida* immunity in the human female genital tract has not been clearly

Address correspondence to Anna-Ursula Happel, anna.happel@uct.ac.za.

The authors declare no conflict of interest.

established. We show that presence of yeast hyphae was associated with increases in Th17 cell-related cytokines and the absence of microbial dysbiosis, and that persistent *Candida* colonization resulted in significant increases in Th17-related cytokines and highly activated Th17-like cell frequencies. Our results suggest that Th17 cells are important for anti-*Candida* immunity in the human female genital tract and that prolonged vaginal *Candida* colonization may contribute to increased HIV risk in Sub-Saharan African adolescents by increasing HIV target cell frequencies and activation.

**KEYWORDS** vaginal candidiasis, Sub-Saharan Africa, genital inflammation, HIV target cells, mucosal immunity

Young women in Sub-Saharan Africa are disproportionally affected by HIV (1) and observational studies have implicated certain hormonal contraceptive types (2) and potentially colonization with vulvovaginal *Candida* (evident by Gram stain) (3) with increased HIV risk in women. In addition, cervicovaginal inflammation has been associated with an increased risk of HIV acquisition (4) through increased numbers and activation of specific cervical HIV target cell phenotypes (5). The major drivers of cervicovaginal inflammation include colonization with nonoptimal bacterial microbiota (6) and the presence of undiagnosed or asymptomatic sexually transmitted infections (STIs) (7). The relationship between these cervicovaginal factors and *Candida* colonization or vulvovaginal candidiasis (VVC) has not been described. Furthermore, it has not been investigated whether hormonal contraceptive type influences incidence of *Candida* colonization or VVC in adolescents at a high risk of HIV acquisition.

The majority of women experience asymptomatic vaginal *Candida* colonization at some point during their lives (8, 9), and it is thought that vaginal defense mechanisms against *Candida* spp. allow for their persistence as commensals (8, 10). When anti-fungal host defenses do not respond adequately or changes in the host environment occur, opportunistic *Candida* spp. can transform from a symptomless colonization into a pathogenic infection, which is accompanied by the presence of budding yeasts and a morphological transition from yeast cells to hyphae, which may result in symptomatic VVC (10). VVC is defined by clinical signs and symptoms of vulvovaginal inflammation, with the presence of budding yeasts, hyphae, or pseudohyphae on a wet mount or Gram stain (8, 9). Interleukin (IL)-17-mediated responses have been shown to be essential in controlling clinical cases of candidiasis at the oral and dermal mucosae (11), although its role in anti-*Candida* immunity in the human female genital tract (FGT) has not been clearly established. Studies in some mouse models have suggested that IL-17 responses were protective against murine VVC (11, 12). However, other studies have found that mice lacking the IL-17/IL-22 signaling pathway showed no evidence of altered VVC susceptibility or immunopathology (13, 14). An experimental *C. albicans* vaccine administered to mice demonstrated a moderate increased time to VVC recurrence that was associated with vaccine-enhanced T-helper 1 (Th1) and T-helper 17 (Th17) responses (15). Th17 cells mediate protection by producing IL-17A, IL-17F, IL-21, and IL-22 (16), while Th1 cells produce IL-1$\beta$, IL-2, IL-12, tumor necrosis factor (TNF)-$\alpha$, and IFN-$\gamma$ (17), but both are thought to be highly susceptible to HIV infection, especially in their activated state (18, 19). Thus, activation of host defenses against vaginal *Candida* colonization or infection in the human FGT may result in increased genital pro-inflammatory cytokine production and enrichment for Th1/Th17 cells, both of which have been linked to an increased risk of HIV acquisition (4, 18, 19).

FGT inflammation and Th1/Th17 cell phenotype and activation are also influenced by hormonal contraceptive type (20, 21), and previous observational studies have suggested that hormonal contraceptive type may influence *Candida* colonization, with hormonal contraceptive users experiencing higher *Candida* prevalence than non-users (22, 23). The prevalence of asymptomatic vaginal *Candida* colonization and symptomatic VVC has been found to be 2-fold higher in women taking combined oral contraceptives (COC) compared to those using other barrier and hormonal contraceptives

(24, 25). A higher prevalence of *Candida* colonization and VVC has also been found in intrauterine device (IUD) users compared to that in women not using IUDs (26, 27). Further, contraceptive vaginal rings like the NuvaRing (28), a novel etonorgesterol/ethinyl estradiol combined contraceptive ring inserted monthly, have been shown *in vitro* to be easily colonized by *Candida* spp., and there is clinical evidence that NuvaRing use is associated with high incidence of *Candida* colonization (29). However, there are limited data on the effect of hormonal contraceptive type on incidence of *Candida* colonization or hyphae formation in adolescents.

In a longitudinal open-label study that randomized South African adolescents to the injectable norethisterone enanthate (Net-En), COC or NuvaRing, we hypothesized that NuvaRing use is associated with increased vaginal colonization with *Candida* spp. compared to COC and Net-En use in Sub-Saharan adolescents at high risk of HIV acquisition, and that it would consequently be associated with increased genital inflammation, higher frequencies and activation of Th17- and Th1-enriched cells, and nonoptimal vaginal microbiota.

## RESULTS

**Cohort characteristics.** Of the 130 enrolled participants, 94 adolescents (median, 17 years old; interquartile range [IQR], 16 to 18 years) who were randomized to Net-En ($n = 32$), NuvaRing ($n = 30$), and COC ($n = 32$) completed the 16-week follow-up visit and had samples available for *Candida* assessment at the baseline and 16-week follow-up time points. The CONSORT diagram of the number of participants who completed each study visit and provided genital samples has been published previously (20). There were no differences in demographics, vaginal insertion practices, risk behavior, or baseline prevalence of bacterial vaginosis (BV) or STIs by randomization arm, as previously described (20). At baseline, 52 participants (55.3%) had evidence of *Candida* colonization by quantitative PCR (qPCR), while 12 participants (12.8%) had evidence of yeast hyphae by Gram stain, which is indicative of a more pathogenic *Candida* infection. Importantly, none of these adolescents reported any vaginal symptoms. When participants were binarized into 'high' and 'low' *Candida* colonization groups based on the median quantitative PCR cycle threshold ($C_T$) value, all but two participants with hypha formation at baseline showed evidence of high *Candida* colonization (10/12; 83.3%).

Prevalence of STIs and most captured sexual behaviors did not differ significantly between adolescents with evidence of vaginal *Candida* colonization compared to those without at baseline, nor did current contraceptive use or contraceptive randomization arm (Table 1). Adolescents with *Candida* colonization more commonly reported 'almost always' or 'always' using condoms (79.9% versus 55.9%; $P = 0.027$) compared to those without *Candida* colonization, consistent with results of previous literature (10). The prevalence of *N. gonorrheae* was 4-fold higher in adolescents with high *Candida* colonization (7/23; 30.4%) compared to those with low (2/29, 6.9%, $P = 0.035$) or no colonization (2/35, 8.6%, $P = 0.011$) (Table 1). Since STIs are highly inflammatory (7) and may confound the relationship between *Candida* colonization and cervicovaginal cytokines or T-cell phenotypes, we adjusted for their presence in subsequent analyses, alongside Herpes simplex virus (HSV)-2 seropositivity and BV.

**_Candida_ colonization and cervicovaginal cytokines.** We previously used unsupervised partition around medoids (PAM) clustering to binarize participants of this cohort into genital cytokine–high and genital cytokine–low clusters based on their overall cytokine profile (20). At baseline, neither *Candida* colonization nor the presence of yeast hyphae drove overall genital tract cytokine status in these South African adolescents, but BV did (Fig. 1A). About 60% of adolescents had highly elevated genital tract cytokine concentrations, but this did not differ by *Candida* colonization or hyphae presence status (Table 1).

In univariate analyses, women with hypha formation at baseline had higher cervicovaginal concentrations of IL-22 (median, 12.79 pg/mL; IQR, 6.80 to 40.78), a cytokine produced by Th17 cells, compared to those without (median, 6.38 pg/mL; IQR, 4.62 to

**TABLE 1** Biological, demographic, and behavioral baseline characteristics in women with Candida colonization and VVC compared to those without[a]

| Characteristic; N (%)[b] | None (n = 35) | Colonization (n = 52) | P | High colonization (n = 23) | Low colonization (n = 29) | P[c] | None (n = 75) | Hypha formation (n = 12) | P |
|---|---|---|---|---|---|---|---|---|---|
| Age; median (IQR) | 17 (16–18) | 17 (16–18) | 0.471 | 17 (16–18) | 18 (16–18) | 0.767 | 17 (16–18) | 17 (16–18) | 0.678 |
| Species | | | | | | | | | |
| C. trachomatis | 15 (42.9) | 17 (32.7) | 0.461 | 11 (47.8) | 6 (20.7) | 0.082 | 27 (36.0) | 5 (41.7) | 0.956 |
| N. gonorrhoea | 3 (8.6) | 9 (17.3) | 0.400 | 7 (30.4) | 2 (6.9) | **0.026** | 9 (12.0) | 3 (25.0) | 0.446 |
| T. vaginalis | 3 (8.6) | 6 (11.5) | 0.931 | 2 (8.7) | 4 (13.8) | 0.757 | 8 (10.7) | 1 (8.3) | 1.000 |
| M. genitalium | 2 (5.7) | 1 (1.9) | 0.725 | 0 (0.0) | 1 (3.4) | 0.506 | 3 (4.0) | 0 (0.0) | 1.000 |
| BV | | | 0.683 | | | 0.674 | | | **0.038** |
| Negative | 19 (54.3) | 24 (46.2) | | 11 (47.8) | 13 (44.8) | | 33 (44.0) | 10 (83.3) | |
| Intermediate | 2 (5.7) | 5 (9.6) | | 1 (4.3) | 4 (13.8) | | 7 (9.3) | 0 (0.0) | |
| Positive | 14 (40.0) | 23 (44.2) | | 11 (47.8) | 12 (41.4) | | 35 (46.7) | 2 (16.7) | |
| HSV-2 serology | | | 0.657 | | | 0.515 | | | 0.536 |
| Negative | 24 (68.6) | 37 (71.2) | | 15 (65.2) | 22 (75.9) | | 54 (72.0) | 7 (58.3) | |
| Positive | 11 (31.4) | 14 (26.9) | | 8 (34.8) | 6 (20.7) | | 20 (26.7) | 5 (41.7) | |
| Equivocal | 0 (0.0) | 1 (1.9) | | 0 (0.0) | 1 (3.4) | | 1 (1.3) | 0 (0.0) | |
| Vaginal pH; mean (SD) | 4.8 (0.6) | 4.9 (0.5) | 0.351 | 4.9 (0.5) | 4.9 (0.5) | 0.506 | 4.9 (0.5) | 4.7 (0.7) | 0.334 |
| CST distribution[d] | | | 0.435 | | | 0.450 | | | 0.101 |
| CST-I | 10 (29.4) | 10 (19.6) | | 6 (23.1) | 4 (16.0) | | 16 (21.9) | 4 (33.3) | |
| CST-III | 6 (17.6) | 14 (27.5) | | 5 (19.2) | 9 (36.0) | | 15 (20.5) | 5 (41.7) | |
| CST-IV | 18 (52.9) | 27 (52.9) | | 15 (57.7) | 12 (48.0) | | 42 (57.5) | 3 (25.0) | |
| Shannon index; median (IQR) | 1.53 (0.43–2.14) | 1.46 (0.69–2.11) | 0.914 | 1.50 (0.70–2.18) | 1.45 (0.72–1.94) | 0.863 | 1.70 (0.54–2.17) | 0.83 (0.54–1.46) | 0.211 |
| High genital cytokines[e] | 21 (60.0) | 33 (63.5) | 0.892 | 14 (60.9) | 19 (65.5) | 0.866 | 48 (64.0) | 6 (50.0) | 0.591 |
| Current contraceptive use | | | 0.246 | | | 0.123 | | | 0.292 |
| COC | 0 (0.0) | 4 (15.4) | | 4 (15.4) | 0 (0.0) | | 2 (2.7) | 2 (16.7) | |
| DMPA | 7 (20.0) | 8 (16.0) | | 4 (15.4) | 4 (16.7) | | 13 (17.8) | 2 (16.7) | |
| Implanon | 0 (0.0) | 3 (6.0) | | 2 (7.7) | 1 (4.2) | | 3 (4.1) | 0 (0.0) | |
| Net-En | 20 (57.1) | 24 (48.0) | | 11 (42.3) | 13 (54.2) | | 38 (52.1) | 6 (50.0) | |
| None | 8 (22.9) | 11 (22.0) | | 5 (19.2) | 6 (25.0) | | 17 (23.3) | 2 (16.7) | |
| Contraceptive randomization arm | | | 0.830 | | | 0.975 | | | 0.294 |
| COC | 10 (28.6) | 18 (34.6) | | 9 (34.6) | 9 (34.6) | | 24 (32.0) | 4 (33.3) | |
| Net-En | 13 (37.1) | 17 (32.7) | | 9 (34.6) | 8 (30.8) | | 28 (37.3) | 2 (16.7) | |
| NuvaRing | 12 (34.3) | 17 (32.7) | | 8 (30.8) | 9 (34.6) | | 23 (30.7) | 6 (50.0) | |
| Previous pregnancy | 7 (20.0) | 5 (9.8) | 0.306 | 1 (4.5) | 4 (13.8) | 0.261 | 11 (14.9) | 1 (8.3) | 0.876 |
| Vaginal douching | 1 (2.9) | 0 (0.0) | 0.849 | 0 (0.0) | 0 (0.0) | 0.478 | 1 (1.4) | 0 (0.0) | 1.000 |
| Intravaginal practices | | | | | | | | | |
| Wash with water | 6 (17.1) | 7 (13.7) | 0.898 | 2 (8.7) | 5 (17.9) | 0.602 | 12 (16.2) | 1 (8.3) | 0.785 |
| Insert herbs | 1 (2.9) | 0 (0.0) | 0.849 | 0 (0.0) | 0 (0.0) | 0.478 | 1 (1.4) | 0 (0.0) | 1.000 |
| Insert cloth | 2 (5.7) | 2 (3.9) | 1.000 | 1 (4.3) | 1 (3.6) | 0.920 | 4 (5.4) | 0 (0.0) | 0.932 |
| Insert medication | 2 (5.7) | 1 (2.0) | 0.738 | 0 (0.0) | 1 (3.6) | 0.510 | 3 (4.1) | 0 (0.0) | 1.000 |
| Tampon use | 2 (5.7) | 4 (7.8) | 1.000 | 1 (4.3) | 3 (10.7) | 0.627 | 6 (8.1) | 0 (0.0) | 0.680 |

**TABLE 1** (Continued)

| Characteristic; N (%)[b] | None (n = 35) | Colonization (n = 52) | P | High colonization (n = 23) | Low colonization (n = 29) | P[c] | None (n = 75) | Hypha formation (n = 12) | P |
|---|---|---|---|---|---|---|---|---|---|
| Current sexual partner | 34 (100.0) | 47 (90.4) | 0.164 | 21 (91.3) | 26 (89.7) | 0.171 | 69 (93.2) | 12 (100.0) | 0.793 |
| General condom use | | | 0.068 | | | 0.218 | | | 0.958 |
| Never | 3 (8.8) | 6 (11.5) | | 2 (8.7) | 4 (13.8) | | 7 (9.5) | 2 (16.7) | |
| Almost never | 5 (14.7) | 4 (7.7) | | 2 (8.7) | 2 (6.9) | | 8 (10.8) | 1 (8.3) | |
| Not sure | 7 (20.6) | 2 (3.8) | | 1 (4.3) | 1 (3.4) | | 8 (10.8) | 1 (8.3) | |
| Almost always | 8 (23.5) | 22 (42.3) | | 8 (34.8) | 14 (48.3) | | 26 (35.1) | 4 (33.3) | |
| Always | 11 (32.4) | 18 (34.6) | | 10 (43.5) | 8 (27.6) | | 25 (33.8) | 4 (33.3) | |
| Sex with ≥5 yrs older partner | | | 0.379 | | | 0.325 | | | 0.242 |
| No | 9 (26.5) | 24 (46.2) | | 10 (43.5) | 14 (48.3) | | 27 (36.5) | 6 (50.0) | 0.242 |
| Don't think so | 3 (8.8) | 3 (5.8) | | 1 (4.3) | 2 (6.9) | | 6 8.1 | 0 (0.0) | |
| Not sure | 14 (41.2) | 15 (28.8) | | 10 (43.5) | 5 (17.2) | | 23 (31.1) | 6 (50.0) | |
| I think so | 2 (5.9) | 1 (1.9) | | 0 (0.0) | 1 (3.4) | | 3 (4.1) | 0 (0.0) | |
| Yes | 6 (17.6) | 9 (17.3) | | 2 (8.7) | 7 (24.1) | | 15 (20.3) | 0 (0.0) | |

[a]BV, bacterial vaginosis; COC, combined oral contraceptives; IQR, interquartile range; SD, standard deviation; CST, community state type.
[b]Unless otherwise specified.
[c]Comparing no, low, and high colonization. Values in bold font are statistically significant.
[d]Based on vaginal bacterial microbiota assessment by 16S rRNA gene sequencing (Balle et al. [20]). CST-I = *L. crispatus*-dominated, low diversity; CST-III = *L. iners*-dominated, low diversity; CST-IV = diverse groups of anaerobic BV-associated bacteria, high diversity.
[e]Based on genital cytokine measurements by Luminex (Konstantinus et al. [21]; Balle et al. [20]). Unsupervised PAM clustering of cytokine concentrations was used to binarize overall cervicovaginal cytokines levels into 'cervicovaginal cytokines–high' and 'cervicovaginal cytokines –low' clusters.

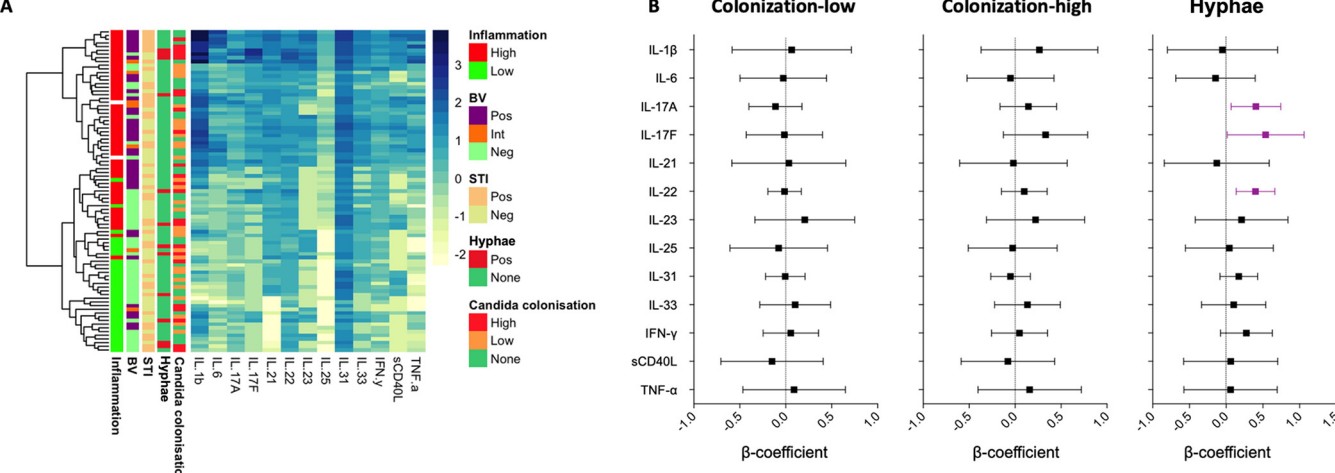

**FIG 1** *Candida* colonization cervicovaginal cytokines at baseline. (A) Heatmap of $\log_{10}$ cytokine concentrations at baseline annotated by cytokine/inflammatory group, bacterial vaginosis (BV) status by Nugent Scoring, presence of bacterial sexually transmitted infections (STIs), yeast hyphae (hyphae) by Gram stain, and *Candida* colonization by qPCR. (B) Multivariate linear regressions showing the association between genital cytokine concentrations in women with low or high *Candida* colonization compared to those without any colonization, and in women with hyphae compared to those without. Each association is shown as a $\beta$-coefficient and the error bars represent the 95% confidence interval (CI). Associations shown in purple are statistically significant. *P* values of ≤0.05 were considered statistically significant.

11.07; *P* = 0.034), while no differences were evident for any other cytokine measured. No differences were noted in cervicovaginal cytokines in women stratified by *Candida* colonization status. As coinfections may influence the relationship between *Candida* colonization and/or hypha formation and genital immunity, we performed multivariate linear regressions adjusting for BV, bacterial STIs, and HSV-2 serology. After adjusting for these confounders, women with hypha formation, but not those with *Candida* colonization only, had significantly higher levels of IL-17A ($\beta$ = 0.41, *P* = 0.019), IL-17F ($\beta$ = 0.54, *P* = 0.044) and IL-22 ($\beta$ = 0.40, *P* = 0.0003) compared to those without, with IL-22 remaining statistically significant after adjusting for multiple comparisons (adjusted *P* = 0.039; Fig. 1B). These data suggest that hypha formation is associated with increased levels of Th17-produced cervicovaginal cytokines, even in the absence of symptoms.

***Candida* colonization and cervical immune cell activation.** We measured cervical cytobrush-derived T-cell frequencies and activation, with a particular emphasis on HIV target cells and those thought to be important for mucosal anti-*Candida* host defense (Fig. S1 in the supplemental material). Since cervical cytobrush samples yield too few CD3+ T cells for intracellular cytokine or transcription factor staining, we used chemokine receptor expression as a proxy for T-helper cell phenotyping. We previously reported a higher frequency of cervical CD3+ CD4+ T cells that were CCR6+ CCR10– (considered Th17-like cells; [21]) compared those that were CCR6– CCR10– (considered Th1/2-enriched cells; [21]) in this cohort. At baseline, the frequency of activated HLA-DR+ CD4+ T cells was lower among adolescents with *Candida* colonization compared to those without (*P* = 0.018), while no differences in CCR5+ CD4+ HIV target cells were observed (Table S1). These adolescents also tended to have lower frequencies of activated, HIV-susceptible Th17-like cells (HLA-DR+, *P* = 0.029; CCR5+, HLA-DR+, *P* = 0.061). On the other hand, adolescents with *Candida* colonization had higher frequencies of CCR6– CCR10– CD4+ T cells (*P* = 0.041), although the frequencies of activated, HIV-susceptible CCR6– CCR10– T cells were lower (CCR5+ HLA-DR +, *P* = 0.047; HLA-DR+, *P* = 0.049) than those in adolescents without *Candida* colonization (Fig. 2A). Similar trends were seen when comparing those with hyphae versus without hyphae on a Gram stain (Table S1 in the supplemental material). After adjusting for BV, bacterial STIs, and HSV-2 serology, women with low *Candida* colonization had significantly higher frequencies of CD4+ T cells ($\beta$ = 12.87, *P* = 0.027) and CCR6– CCR10– T cells ($\beta$ = 10.56, *P* = 0.019) but lower frequencies of Th17-like cells ($\beta$ = −10.27, *P* = 0.026)

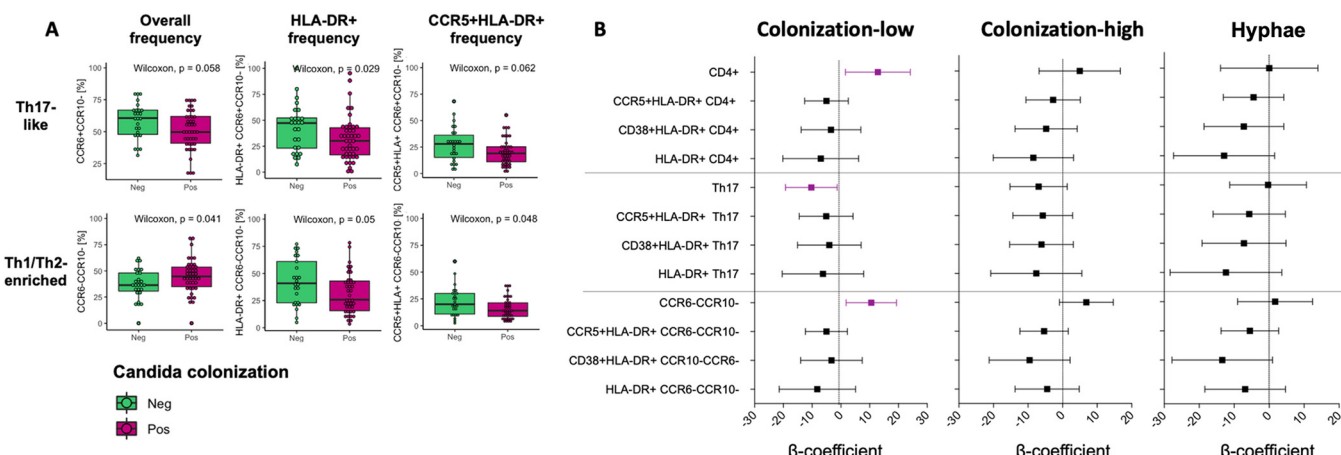

**FIG 2** *Candida* colonization and cervical immune cells at baseline. (A) Cervical Th17 cells (CCR6+ CCR10−) and Th1/Th2-enriched cells (CCR6− CCR10−) were identified by expression of CCR6 and CCR10 on CD4+ T cells. The frequencies of Th17-like and Th1/Th2-enriched cells and expression of HLA-DR and CCR5 on these cells in women with *Candida* colonization (purple) and in those without (green) was assessed at the baseline, and Wilcoxon rank-sum tests were applied to compare the frequencies between the groups. *P* values of ≤0.05 were considered significant. (B) Multivariate linear regressions showing association between T-cell phenotypes in women with low or high *Candida* colonization compared to that in those without any colonization and in women with hyphae compared to in those without. Each association is shown as a *β*-coefficient and error bars represent the 95% CI. Associations shown in purple were statistically significant prior to adjusting for multiple comparisons. *P* values of ≤0.05 were considered statistically significant.

compared to women without *Candida* colonization, and similar trends seen in women with high *Candida* colonization and those who had hyphae present compared to those without (Fig. 2B). These data suggest that *Candida* colonization, independent of severity and symptoms, is associated with higher frequencies of some HIV target cells (CCR6− CCR10− CD4+ T cells), although overall frequencies of Th17-like cells were lower with *Candida* colonization.

**Candida colonization and vaginal microbiota.** At baseline, BV was significantly less prevalent in adolescents with hyphae (2/12, 16.7%; Nugent 7 to 10) compared to adolescents without hyphae on Gram stains of cervicovaginal secretions (42/75, 56.0%; *P* = 0.012) (Table 1). Consistent with this, significantly more of the participants with hyphae had a vaginal microbiota dominated by *Lactobacillus* spp., particularly *L. iners* (community state type [CST]-III; 5/12, 41.7%) and *L. crispatus* (CST-I; 4/12, 33.3%) compared to those without (15/75, 20.5% and 16/75, 21.9%, respectively; *P* = 0.030) (Fig. 3A). The Shannon index for the cervicovaginal microbiota, a measure of bacterial species diversity within a community, tended to be lower among women with hypha formation (median, 0.83; IQR, 0.54 to 1.46) compared to those without (median, 1.70; IQR, 0.54 to 2.17; *P* = 0.211) (Fig. 3A). We used DESeq2 to identify differentially abundant taxa between adolescents with *Candida* colonization or hypha formation and those without (Fig. 3B). *Prevotella intermedia* and *Fusobacterium nucleatum* were significantly less abundant in vaginal specimens from adolescents with *Candida* colonization compared to in specimens from those without *Candida*. In addition to *Prevotella* and *Fusobacterium*, participants with hypha formation had lower relative abundances of taxa belonging to the *Megasphaera*, *Mobiluncus*, *Moryella*, *Fusobacterium*, *Porphyromonas*, *Campylobacter*, and *Gardnerella* genera than those without (Fig. 3B, Table S2). Overall, these data suggest that there may be an association between absence of BV-associated bacteria and hypha formation in this cohort.

**Distinguishing adolescents with yeast hyphae using cytokine, bacteria, and Th17-like cell signatures.** Since we identified associations between the presence of hyphae and specific cytokines, vaginal bacteria, and cervical T-cell phenotypes, all of which may be related to each other, we integrated baseline cervicovaginal cytokines, Th17-like immune cell phenotypes, and microbial taxa at the genus level to identify the genital biomarkers that were most distinct between adolescents with hypha formation and those without. Using DIABLO (30), which integrates multiple data sets using supervised analysis, we found that the variables which explained the most variance between adolescents with hypha formation and those without included five cytokines (sCD40L, IL-

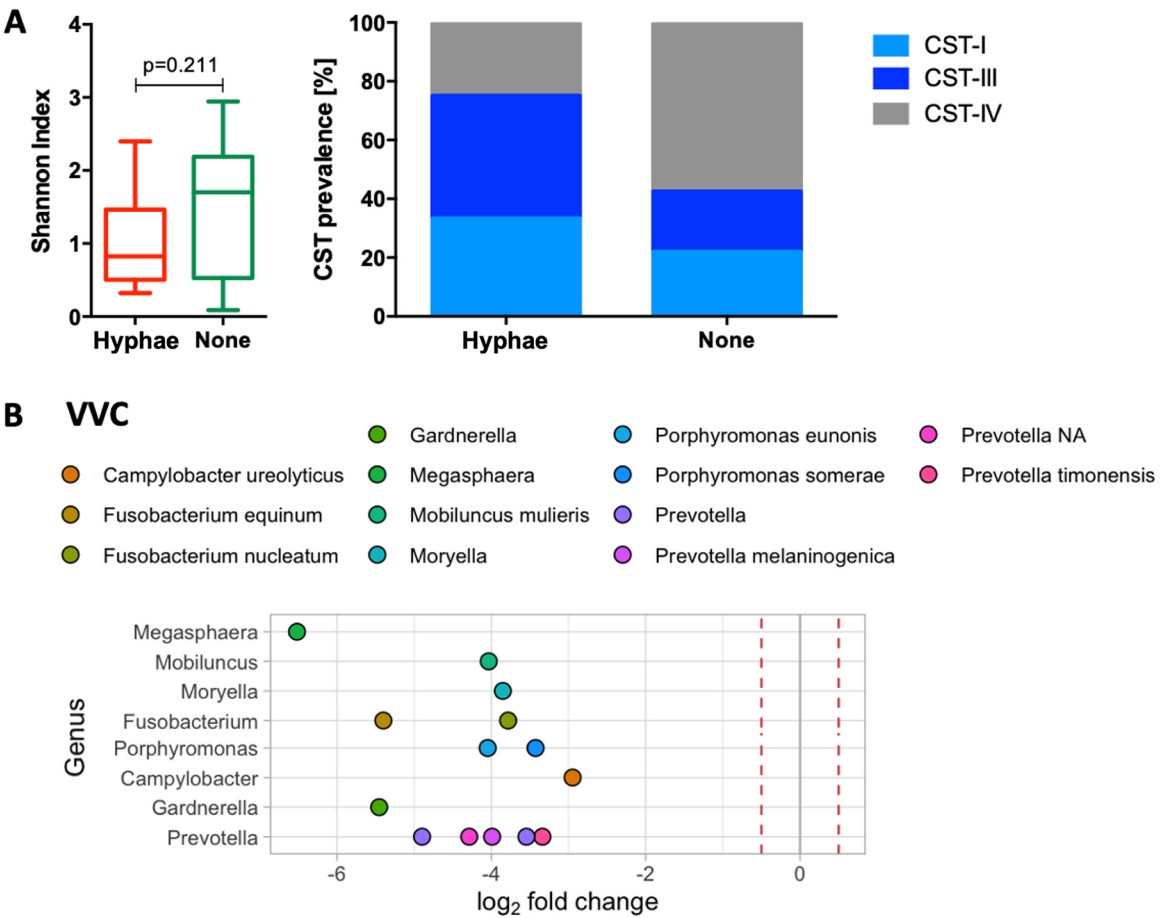

**FIG 3** Yeast hyphae and vaginal microbiota at baseline. (A) The Shannon index, a measure of within-sample species diversity, was compared in women with yeast hyphae and in those without at baseline using a Wilcoxon rank-sum test. Prevalence of community state type (CST)-I (*Lactobacillus crispatus*-dominated), CST-III (*L. iners*-dominated) and CST-IV (dominated by diverse, bacterial vaginosis [BV]-associated bacteria) in women with hyphae and in those without. (B) Fold-change differences in the abundances of specific vaginal bacterial taxa in women with hyphae compared to in those without. Dots to the left of 0 represent a fold-change decrease. Red dotted lines represent a 0.5-fold change difference in abundance. Only differential abundances where $P \leq 0.01$ after adjusting for multiple comparisons were included.

17F, IL-22, IL-21, and IL-6), five bacterial genera (*Lactobacillus, Fusobacterium, Moryella, Porphyromonas,* and *Campylobacter*) and multiple highly activated Th17-like cell phenotypes (CD38+, CCR5+, CD38+ CCR5+, CD38+ HLA-DR+, CCR5+ HLA-DR+, CCR5+ CD38+ HLA-DR+) (Fig. 4A). The loading weights of the selected variables suggest that differences among women with hypha formation were primarily driven by higher concentrations of IL-22 (a member of the IL-10 superfamily) but lower IL-21 and IL-6 concentrations, and lower *Porphyromonas* and *Campylobacter* abundance (Fig. 4). In agreement with our previous analyses, adolescents with hypha formation had higher levels of IL-17F, IL-22, and *Lactobacillus* but lower levels of other selected bacterial and immune cell features compared to those without hyphae (Fig. 4B). *Lactobacillus* relative abundance was negatively correlated with cytokine and Th17 cell levels, while positive correlations were observed between the other selected bacteria, cytokines, and Th17 cell phenotypes (Fig. 4B). After accounting for the relationship between cervicovaginal cytokines, bacteria, and Th17-like cells, the most distinguishing features between adolescents with and without hypha formation were increased levels of IL-22, decreased levels of IL-6, and decreased abundance of *Campylobacter ureolyticus*. However, these selected features poorly distinguished women with hyphae from those without, but our sample size is small (Fig. 4C).

**Hormonal contraceptive use and *Candida* colonization.** Our data suggest that

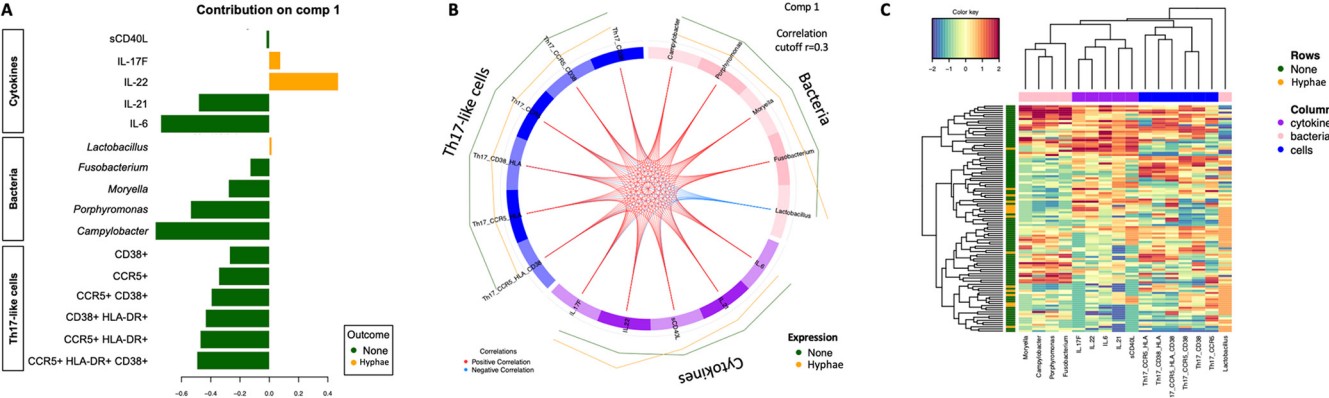

**FIG 4** Distinguishing women with yeast hyphae from those without using cytokine, bacteria, and Th17 cell signatures. Multiomics feature selection model including all cytokines, bacterial taxa merged at the genus level, and Th17-like cell phenotypes which explain the highest variance when comparing women with yeast hyphae to those without. (A) Loading plot showing the contributions of the different selected biomarkers. Color indicates the group in which the variable has the maximum level of expression, using the median. (B) Circos plot showing the expression of the selected biomarkers for women without (outer green line) and those with hyphae (outer orange line), and correlations among the selected biomarkers (inner red and blue lines). Only associations above the threshold correlation cutoff, 0.3, were included. (C) Multiomics feature expression for each participant comparing the integrated cytokine, microbial, and Th17 phenotype profiles of women with hyphae (orange) and of those without (green). Cytokine (purple), bacteria (pink), and Th17-like cell (blue) clusters are shown on the top of the heatmap, and only selected features were included. Color key shows the range of correlation values.

*Candida* colonization is associated with higher frequencies of CCR6− CCR10− CD4$^+$ T cells, a population that previously has been described as being enriched for Th1/Th2 cells (21), and that hypha formation is associated with significantly increased concentrations of Th17-produced cytokines. Because Th1 and Th17 cell phenotypes are both highly susceptible to HIV infection (18, 19), the question arises of whether the use of certain hormonal contraceptive types would result in altered incidence of *Candida* colonization or hyphae. No differences in the prevalence of *Candida* colonization by hormonal contraceptive type were observed between that at the matched baseline and that at 16 weeks post-randomization (Fig. 5A). The presence of hyphae increased 3-fold in participants randomized to Net-En from 6.7% at the baseline to 20.7% at 16

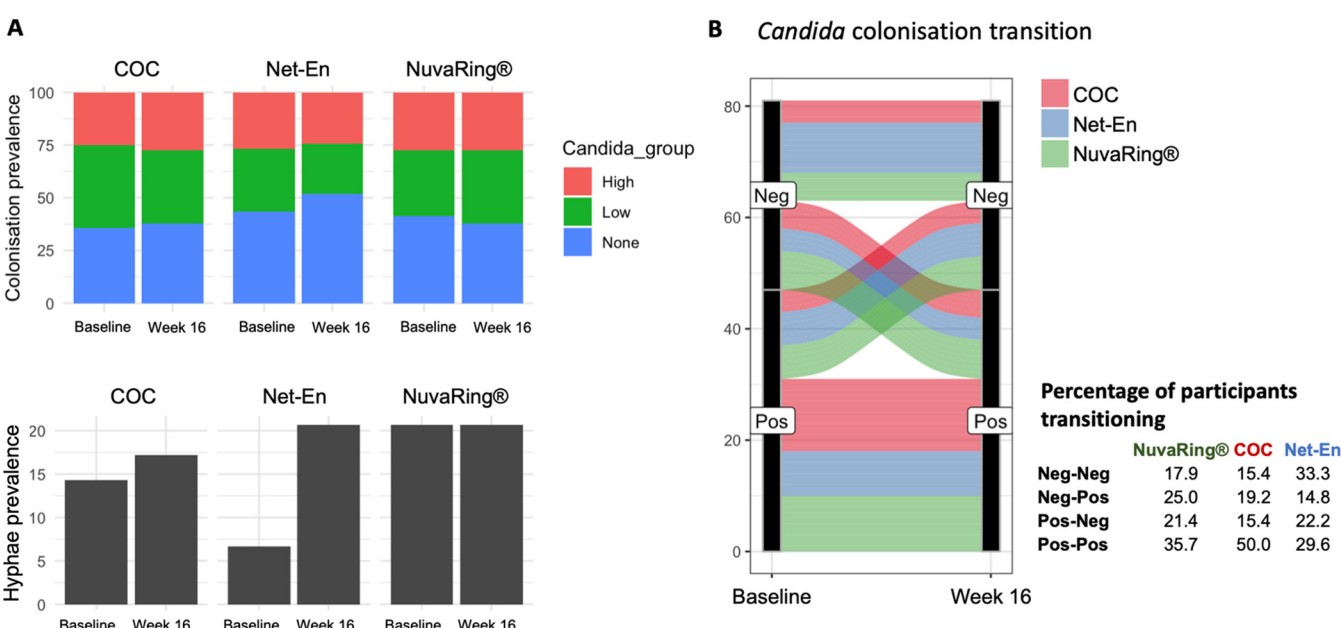

**FIG 5** Hormonal contraceptive use and *Candida* colonization. (A) Prevalence of high, low, or no *Candida* colonization or hyphae at the baseline and at the 16-week follow-up in women randomized to combined oral contraceptives (COC), injectable Net-En, and NuvaRing. (B) Changes in *Candida* status of participants who were colonized with *Candida* (Pos) or *Candida*-free (Neg) from the baseline to the 16-week follow-up. Each line presents one participant and is color-coded by contraceptive randomization group.

weeks post-randomization ($P = 0.116$), while the prevalence in adolescents randomized to COC and NuvaRing remained unchanged (Fig. 5A). No differences in the percentage of participants experiencing changes in *Candida* colonization from the baseline to week 16 were observed by hormonal contraceptive type (Fig. 5B). In an intention-to-treat analysis, the incidence rate (IR) of hypha formation was 0.51 cases per person-years (95% confidence interval [CI], 0.28 to 0.94) in adolescence randomized to the Net-En arm, 0.41 cases per person-years (95% CI, 0.19 to 0.87) in the COC arm, and 0.32 cases per person-years (95% CI, 0.13 to 0.83) in the NuvaRing arm. Similarly, the proportion of adolescents in the NuvaRing arm who acquired *Candida* colonization from baseline to follow-up was almost 2-fold higher than that of those randomized to Net-En (7/12; 58.3% versus 4/13; 30.8%), with an IR for *Candida* colonization of 0.41 cases per person-years (95% CI, 0.19 to 0.87) in the Net-En arm, 0.51 cases per person-years (95% CI, 0.28 to 0.94) in the COC arm and 0.76 cases per person-years (95% CI, 0.53 to 1.10) in the NuvaRing arm. Overall, these observations suggest that NuvaRing, Net-En, and COC use had little influence on *Candida* colonization or hypha formation in this cohort with a small number of participants.

**Effect of *Candida* persistence, clearance, and acquisition on cervicovaginal immunity.** In a longitudinal analysis, there were 17 adolescents who remained free of *Candida* throughout the 16-week period (assessed by qPCR), 15 who cleared their *Candida* colonization, 16 who acquired *Candida* colonization, and 31 who remained persistently colonized.

In a paired analysis, participants who remained colonized with *Candida* through time (Pos-Pos) generally experienced an increase in the concentrations of cervicovaginal cytokines produced by Th17 cells (Fig. 6A), and in those important for Th17 cell differentiation (Fig. 6B) and Th17 cell regulation (Fig. 6C). Concentrations of IL-6, IL-1$\beta$, TNF-$\alpha$, and IL-31 (Fig. 6B) increased significantly from the baseline to the 16-week follow-up time point, with IL-6 remaining statistically significant after multiple-comparison adjustment (adjusted $P = 0.0486$). In contrast, participants who remained free of *Candida* throughout the 16-week study period (Neg-Neg) experienced an overall decrease in cervicovaginal cytokines, particularly sCD40L, IL-22, IL-25, and IL-31, with sCD40L remaining statistically significant after multiple-comparison adjustment (adjusted $P = 0.0271$). In contrast, no change in cervicovaginal cytokines measured was observed for participants who cleared (Pos-Neg) or acquired (Neg-Pos) *Candida* colonization between baseline and week 16 (Fig. 6A–C). Overall, these results suggest that persistent *Candida* colonization may be associated with a Th17 response in the FGT, involving cytokines which are thought to be important for Th17 cell differentiation (IL-1$\beta$, IL-6, TNF-$\alpha$) and regulation (IL-31). Further, significant increases in frequencies of highly activated HLA-DR+ (adjusted $P = 0.0144$) and CCR5+CD38+HLA-DR+ (adjusted $P = 0.0018$) Th17-like cells (Fig. 7A) and highly activated HLA-DR+ (adjusted $P = 0.0240$) and CCR5+CD38+HLA-DR+ (adjusted $P = 0.0132$) CCR6-CCR10- CD4$^+$ T cells (Fig. 7B) were observed from baseline to the 16-week visit in participants remaining colonized with *Candida* (Pos-Pos), while again no changes in frequencies of these CD4$^+$ T cell phenotypes were observed for participants clearing or acquiring *Candida* colonization. Of the 31 participants who remained colonized with *Candida*, the majority (20/31; 64.5%) did not experience a change in colonization severity, while about a quarter (8/31; 25.8%) progressed from low to high *Candida* colonization, and few (3/31; 9.7%) moved from high to low colonization, suggesting that an increase in *Candida* colonization severity or symptoms is not required for increases in cytokine concentrations and T cell activation, but persistent colonization is sufficient. We explored associations between *Candida* clearance or acquisition and cervicovaginal cytokine concentrations (Fig. 6D) and T cell phenotypes (Fig. 7C) in more detail using multivariate linear regressions. After adjusting for randomization arm, HSV-2 serology, BV, and bacterial STIs at baseline to account for antibiotic exposure, participants who cleared *Candida* colonization experienced a decrease in IL-17F ($\beta= -0.90$; 95% CI, $-1.8$ to 0; $P = 0.05$) compared to those remaining colonized, while adolescents acquiring *Candida* colonization experienced a decrease in IL-17A ($\beta= -0.60$;

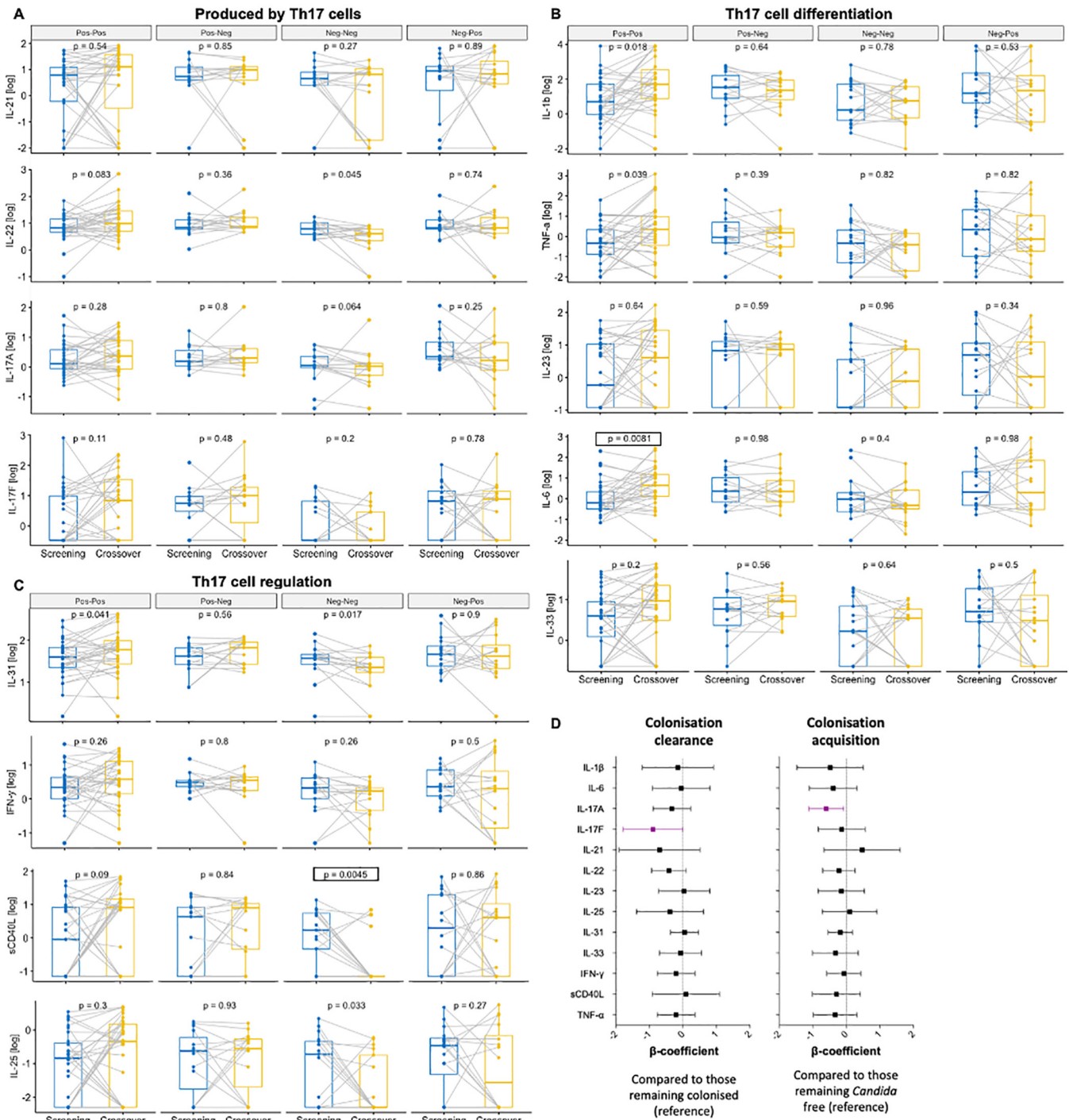

**FIG 6** Persistence, clearance, and acquisition of *Candida* colonization from baseline to 16 weeks, and cervicovaginal cytokines. Paired concentrations of cytokines produced by Th17 cells (A), those important for Th17 cell differentiation (B), and those for Th17 cell regulation (C) from baseline to 16 weeks in women who remained persistently colonized with *Candida* (Pos-Pos), those who cleared *Candida* (Pos-Neg), those who remained free of *Candida* (Neg-Neg), and those who acquired *Candida* (Neg-Pos). *P* values prior to multiple-comparison adjustment are displayed. Those remaining statistically significant after multiple-comparison adjustment are indicated by a box. (D) Multivariate linear regressions showing the association between cervicovaginal cytokines and *Candida* clearance (compared to women remaining colonized) or acquisition (compared to women remaining non-colonized). Each association is shown as a $\beta$-coefficient and error bars represent the 95% CI. Associations shown purple were statistically significant prior to adjusting for multiple comparisons. *P* values of ≤0.05 were considered statistically significant.

95% CI, −1.12 to −0.070; *P* = 0.027) compared to those who remained *Candida*-free, neither of which remained significant after accounting for multiple comparison (Fig. 6D). In contrast, no significant changes were observed in immune cell frequencies or activation in multivariate linear regressions of participants who cleared or acquired *Candida*

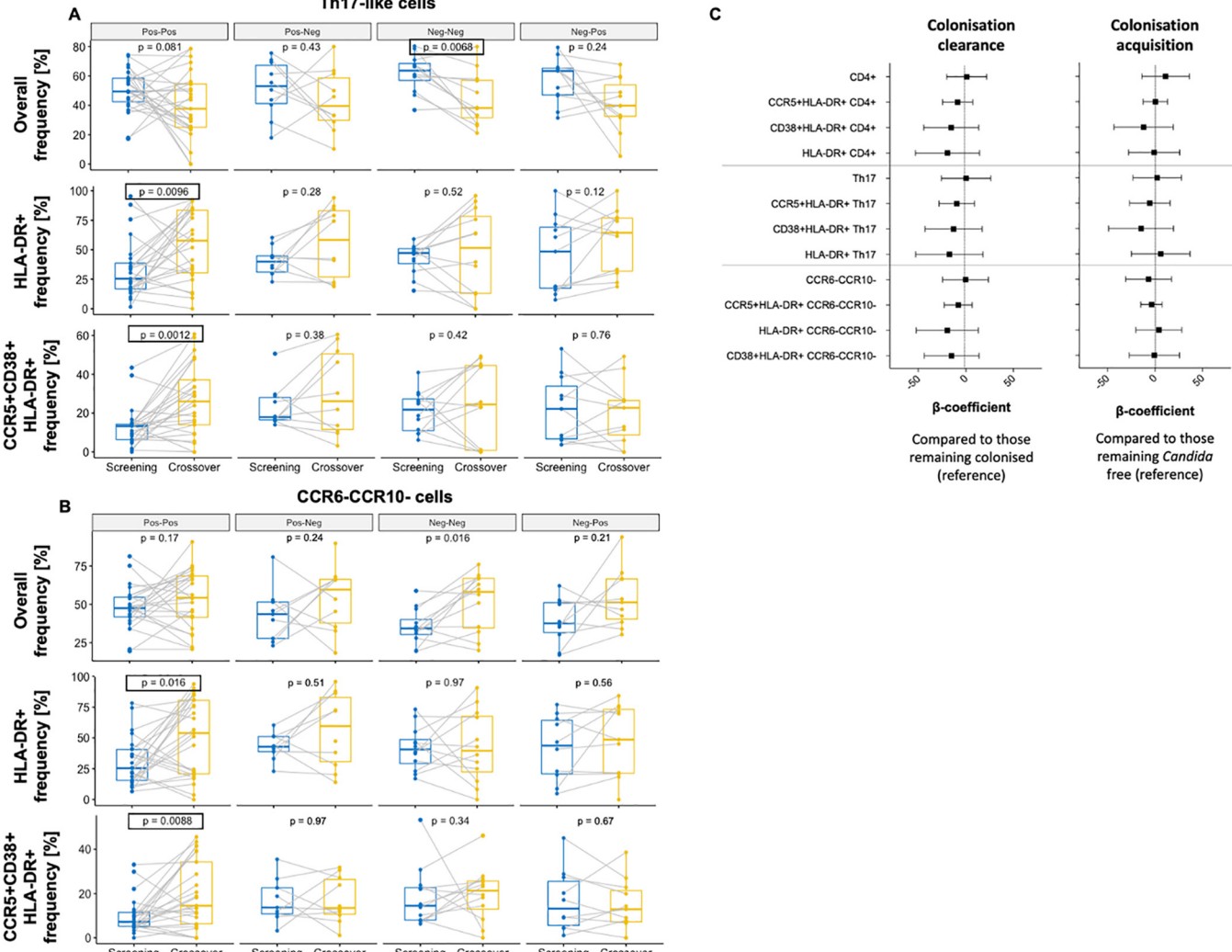

**FIG 7** Persistence, clearance, and acquisition of *Candida* colonization from baseline to 16 weeks and cervical immune cell phenotype. Paired frequencies of Th17-like (A) and CCR6– CCR10– CD4+ T cells (B) from baseline to 16 weeks in women who remained persistently colonized with *Candida* (Pos-Pos), those who cleared *Candida* (Pos-Neg), those who remained free of *Candida* (Neg-Neg), and those who acquired *Candida* (Neg-Pos). *P* values prior to multiple comparison adjustment are shown. Those remaining statistically significant after multiple-comparison adjustment are indicated by a box. (C) Multivariate linear regressions showing the association between cervical T-cell phenotypes and *Candida* clearance (compared to women remaining colonized) or acquisition (compared to women remaining non-colonized). Each association is shown as a *β*-coefficient and error bars represent the 95% CI. *P* values of ≤0.05 were considered statistically significant.

(Fig. 7C). In summary, these data indicate that prolonged *Candida* colonization, even when asymptomatic, is associated with a Th17 immune response in the human FGT, while no effects of clearance or acquisitions were measurable shortly after these events.

## DISCUSSION

Globally, adolescent girls and young women in Sub-Saharan Africa are at the highest risk for HIV infection (31). Cervicovaginal cytokines, T-cell phenotype and activation, and microbiota all influence sexual health and the risk of adverse outcomes in women, and may be influenced by hormonal contraceptive type. Cervical T-cell phenotype and/or activation (18, 19), genital tract inflammation (4, 6), and cervicovaginal microbiota also influence women's risk of HIV infection (6), and STIs, even when asymptomatic, are associated with increased genital tract inflammation and HIV risk (32, 33). We examined whether *Candida* colonization and/or hyphae presence is associated with cervicovaginal cytokine concentrations, T-cell phenotype and activation, and vaginal microbiota composition in adolescents not reporting any symptoms. Here, we

present data suggesting that *Candida* hyphae induce IL-17- and IL-22-mediated responses in the FGT of adolescents. Both of these cytokines have been shown to be produced by a range of mucosal immune cells, including various innate lymphoid cells, such as natural killer (NK), NK T, or lymphoid tissue-inducer cells, activated CD4$^+$ and CD8$^+$ T cells, mucosa-associated invariant T cells, gamma delta T cells, and Th22 and Th17 cells (16, 34, 35). As expected, we found higher frequencies of CCR6– CCR10– CD4$^+$ T cells (a cell subset previously described as Th1/Th2- enriched) in adolescents with *Candida* colonization, yet lower frequencies of activated CCR6–CCR10– CD4$^+$ T cells and CCR6+ CCR10– Th17-like cells. After activation, T cells undergo a clonal expansion and differentiation, a burst of cytokine production, and finally extensive apoptosis, a phenomenon referred to as activation-induced cell death, which has also been described for both Th1 and Th17 cells (36, 37). It therefore seems plausible that high concentrations of IL-17 and IL-22 cause Th17-cell activation, after which the activated cells undergo cell death, which might explain the observed lower frequencies of activated CCR6–CCR10– and CCR6+ CCR10– T cells. Alternatively, the activated cells might have been more susceptible to cell death during sample collecting and processing for flow cytometry. Together, our study provides preliminary evidence that Th1/Th17-mediated immunity might be important in controlling vaginal *Candida* colonization and hypha formation in humans. Prior to this study, this had not been investigated in humans and evidence from mouse models was inconclusive (11–15).

Related to this, we found that persistent *Candida* colonization over 16 weeks of follow-up, even when subclinical, led to increased frequencies of highly activated CD4$^+$ T cells, including Th17-like and CCR6– CCR10– cells, alongside increased levels of cytokines related to Th17 cell differentiation and regulation. For other FGT pathogens, such as human papillomaviruses or *C. trachomatis*, it has been suggested that persistent infection is associated with severe adverse health outcomes, while acquisition and clearance are common and have limited adverse effects (38, 39). Whether this is similarly the case for vaginal *Candida* colonization remains to be determined. Furthermore, whether persistent vaginal *Candida* colonization is associated with subsequent changes in the integrity and permeability of the vaginal barrier, as described for *Candida* colonization at the intestinal and oral mucosa (40, 41), which may subsequently influence HIV risk (42), should be investigated.

Given previously reported associations between HIV risk and activated Th1/Th17 cell frequencies (18, 19), genital tract inflammation (4, 43), and vaginal epithelial barrier disruption (42), these findings raise the possibility that persistent *Candida* colonization could increase the risk of HIV acquisition even in the absence of symptoms. Chronic colonization with *Candida* at other mucosal surfaces has been associated with a dysregulated production of cytokines (44). If persistent vaginal *Candida* colonization indeed leads to increased production of Th17-related cytokines and increased frequencies of T-cell phenotypes that are susceptible to HIV infection, this may impact the risk of HIV acquisition in this key population, and screening for *Candida* colonization and subsequent treatment may mitigate some of the risk. These findings are preliminary, but important, and they require confirmation in a larger, independent cohort as well as assessment of a causal relationship between persistent *Candida* colonization and cervicovaginal immunity and/or risk of HIV acquisition, using humanized animal models or longitudinal clinical trials with HIV acquisition as an endpoint to provide conclusive evidence. Whether persistent *Candida* colonization itself, or other biological and/or behavioral factors associated with prolonged *Candida* colonization, led to the observed effects on cervicovaginal immunity needs to be investigated.

The inverse association between hypha formation and BV might be surprising, as *in vitro* studies have suggested protective roles of *Lactobacillus* spp. and their metabolites against mucosal *Candida* infections (45). However, previous clinical studies have also described that BV diagnosed by Nugent score is associated with an absence of vaginal yeast (46, 47). Similar to our findings, recent 16S rRNA gene sequencing-based studies have found that higher relative abundances of *Megasphaera* spp. type 1 and 2 and

*Mageeibacillus indolicus* were associated with a lower risk of detecting vulvovaginal yeast (48), and that the odds of detecting *C. albicans* were 2-folds higher in women with *L. crispatus*-dominated vaginal CSTs compared to CSTs with lower *Lactobacillus* spp. abundances (49). Collectively, these data suggest that *in vivo*, vaginal *Candida* colonization is negatively associated with a vaginal microbiota composition that is characteristic of BV. The reasons for this observation need to be further investigated, but one proposed mechanistic explanation may be related to the ratio of lactic to acetic acid in the vaginal environment (49). BV-associated bacteria are more sensitive to lactic acid (50), while *C. albicans* is more sensitive to acetic acid (51). BV-associated bacteria produce acetic acid rather than lactic acid (52), which may subsequently inhibit *Candida* growth. Further supporting this theory is the fact that *L. crispatus* strains produce more lactic acid than acetic acid (53), which results in the inhibition of BV-associated bacteria but might have limited effects on *Candida* spp. *in vivo*.

Previous observational studies have reported that certain hormonal contraceptive types are associated with increased *Candida* colonization and/or infection (22–27, 29). We therefore examined this within our randomized trial. Our data did not provide evidence that any contraceptive modality tested in this population of South African adolescents was associated with increased *Candida* colonization and/or hypha formation; however, our sample sizes were small. Although the prevalence of yeast hyphae as observed by Gram stain increased 3-fold from baseline to week 16 in participants randomized to Net-En, but not in COC or NuvaRing users, with the highest incidence of yeast hypha formation seen among Net-En users, this was not statistically significant. A larger, more adequately powered study might have been able to detect significant differences by contraceptive type. Furthermore, a previous study has shown that continuous, rather than intermittent, NuvaRing use is associated with symptomatic *Candida* infection (29). Adolescents in this cohort were given the option to either use the NuvaRing for 3 weeks followed by 7 days of non-use, or use it continuously. Many were not familiar with the NuvaRing prior to enrollment, one-third asked to change to another contraceptive method, and the self-report of perfect adherence was 78% (54). Therefore, some adolescents used the ring intermittently, rather than continuously, which may explain why we did not observe increased *Candida* colonization, hypha formation, and/or symptomatic VVC with NuvaRing use.

The limitations of this study are that the primary outcome of the trial was not to evaluate the effect of contraceptive types on *Candida* colonization and hypha formation incidence and the study was thus not powered to detect differences, and that not all participants were contraceptive-naive prior to enrollment. The sample size of the sub-study was limited to that of the parent study. Thus, the number of adolescents with yeast hyphae was small and no cases of symptomatic VVC occurred in the parent study, which limited our ability to make confident conclusions on the relationship between yeast hyphae and microbial and immunological signatures of the FGT. However, we were able to evaluate relevant associations between asymptomatic *Candida* colonization, inflammation, immune cells, and microbiota. We acknowledge that our flow cytometry panel included surface markers only, which allowed us to detect Th1/Th2- and Th17-enriched T cells; however, these cell populations may have contained small proportions of other cell types. The numbers of cervical cells obtained using cytobrush sampling were limited, and thus measurement of intracellular expression of cytokines after stimulation was not feasible.

In conclusion, this study suggests that Th17-mediated immunity is important to control *Candida* in the human female genital tract, and that persistent colonization with *Candida* may be associated with increases in Th17 cell activation and related cytokines, even when no symptoms are present. These findings have implications for adolescents at high risk of HIV and emphasize the need to evaluate the causal relationship between *Candida*, genital tract cytokines, and T-cell phenotype and function in detail in a larger, independent cohort to better inform sexual health care in Sub-Saharan African women.

## MATERIALS AND METHODS

**Study cohort.** Between 22 September 2015 and 30 June 2017, 130 HIV-seronegative adolescent females, aged 15 to 19 years, were recruited through the UChoose Trial, an open-label, randomized crossover study designed to evaluate the feasibility of different hormonal contraceptive options among adolescents (54). UChoose was approved by the Division of AIDS and the University of Cape Town (UCT) Health Science Research Ethics Committee (HREC 801/2014), was conducted in full compliance with South African Good Clinical Practice (SA-GCP), ICH76 GCP, and ICMJE guidelines, and was registered at ClinicalTrials.gov (https://clinicaltrials.gov/ct2/show/NCT02404038). Participants 18 years or older provided informed consent, while participants younger than 18 years provided assent and informed consent was obtained from a parent or legal guardian. Eligibility criteria are described in detail elsewhere (54), but all adolescents were either contraception-naive or on a short-term contraceptive and were willing to be assigned in a 1:1:1 ratio to one of three study arms for a 16-week period: injectable hormonal contraception (Net-En) once every 8 weeks, a combined contraceptive intravaginal ring (NuvaRing; MSD Pty LTD) to be inserted once every 28 days (and removed after 21 days of each 28-day insertion), or COC (Triphasil or Nordette). Participants returned for randomization within 40 days of the screening visit, and mucosal samples (details below) were collected at baseline and 16 weeks after hormonal contraceptive initiation for all women. Randomization was performed using random number sequence in Stata and provided to the pharmacist in sealed envelopes. All participants in the UChoose Trial were offered enrollment into this mucosal sub-study.

**Power calculations.** The sample size of the sub-study was limited to that of the parent study, where power calculations were based on the trial's primary outcome, which was the relative acceptability of combined contraceptive vaginal ring versus other modalities based on the total score for the ORTHO Birth Control Satisfaction Assessment Tool (ORTHO BC-SAT) questionnaire at 4 months after randomization (54). This sub-study used samples collected as part of the parent trial with the aim of assessing associations between *Candida* colonization/hypha formation and hormonal contraceptive use, genital inflammation, cervical T-cell phenotypes, and microbiota composition.

**Sample collection.** At the baseline and the follow-up at week 16 post-randomization, a rapid HIV and a pregnancy test were performed and, if positive, the participant was referred for clinical care. A detailed interviewer-assisted questionnaire assessing medical history, sexual behavior, menstrual cycle, contraceptive use, intravaginal practices, and antibiotic use was completed. Blood was collected for HSV-2 serology; vulvovaginal swabs were collected for STI, BV, yeast hyphae, pH testing (20, 21), 16S rRNA gene sequencing (20) and *Candida* qPCRs; and menstrual cups were used to collect cervicovaginal secretions for cytokine assessment by Luminex (20, 21). A cervical cytobrush was used to collect mucosal T cells (21). No samples were collected during menstruation; instead, the visit was rescheduled.

**Testing for STIs, BV, and microscopic candidiasis.** Serology for HSV-2 gG2 was performed using an enzyme-linked immunosorbent assay (HerpeSelect, Focus Diagnostics, USA). Vulvovaginal swabs were screened for *Chlamydia trachomatis*, *Neisseria gonorrhoeae*, *Mycoplasma genitalium*, and *Trichomonas vaginalis* by multiplex PCR (20). BV was determined by Nugent scoring, with 7 to 10 being considered BV-positive, 6 to 4 intermediate, and 0 to 3 BV negative. As per the 2015 Sexually Transmitted Diseases Treatment Guidelines from the Centers for Disease Control and Prevention (CDC), yeast presence was evaluated by a Gram stain of vaginal fluid which demonstrated budding yeasts, hyphae, or pseudohyphae (55). Vaginal pH was measured using color-fixed indicator strips (Macherey-Nagel). Laboratory-diagnosed STIs were treated with appropriate target therapy, and treatment for BV was offered to participants presenting with clinical symptoms, as per national guidelines.

**Testing for *Candida* colonization.** Lateral wall vaginal swabs were treated with an enzyme cocktail consisting of mutanolysin (25kU/mL, Sigma-Aldrich), lysozyme (450 kU/mL, Sigma-Aldrich), and lysostaphin (4 kU, Sigma-Aldrich) for 1 h at 37°C. Genomic DNA was extracted using a Quick-DNA Fungal/Bacterial Miniprep Kit (Zymo Research) following the manufacturer's protocol. Mechanical disruption was performed in a Qiagen TissueLyser LT for 5 min at 50 oz. We used a previously validated *Candida*-specific real-time PCR assay to determine the presence of *C. albicans*, *C. glabrata*, *C. tropicalis*, *C. parapsilosis*, *C. guilliermondii*, *C. lusitaniae*, *C. dubliniensis*, and *C. krusei* (56). The ITS-2 variable region of the 28S ribosomal gene was amplified by PCR using the primers 5.8S-1F/28S-1R_Forward (5'-CAA CGG ATC TCT TGG TTC TC-3') and 5.8S-1F/28S-1R_Reverse (5'-CGG GTA GTC CTA CCT GAT TT-3'). For each reaction, 1 $\mu$L of 5.8S-1F/28S-1R_Forward primer (100 $\mu$M), 1 $\mu$L of 5.8S-1F/28S-1R_Reverse primer (100 $\mu$M), 5 $\mu$L PowerUp SYBR Green Master Mix, 2 $\mu$L template DNA (10 to 20 ng/$\mu$L), and 1 $\mu$L molecular-grade water was used, for a final reaction volume of 10 $\mu$L. Real-time PCR was performed using a Applied Biosystems QuantStudio 7 Flex real-time PCR system with the following program: 50°C for 2 min (UDG activation) and 95°C for 2 min (*Taq* activation), followed by 40 cycles of 95°C for 15 s (denaturation) and 60°C for 1 min (annealing and extension). For each PCR run, a panel of DNA samples consisting of 5 pg, 500 fg, 100 fg, and 10 fg of *C. albicans* DNA was included, as well as a no-template control. All samples were run in duplicate. The reported lower limit of detection for this assay was 10 fg for all *Candida* species besides *C. glabrata*, for which it was 100 fg. Thus, only samples for which both reactions had a $C_T$ value above that of the 10 fg *C. albicans* DNA were deemed to be colonized with *Candida*, while those with one value below and another above were deemed to be indeterminate and were not included in subsequent analyses. Samples with measurable *Candida* colonization were further binarized into 'high' and 'low' *Candida* colonization based on the median $C_T$ value measured.

**Assessment of vaginal bacterial microbiota.** As described in detail previously (20), extracted genomic DNA from lateral wall vaginal swabs was used to amplify the V4 hypervariable region of the bacterial 16S rRNA gene, using 515 F and 806 R primers. Amplicons from 96 samples were pooled in equal mass amounts and the libraries were sequenced on an Illumina MiSeq platform (300-bp paired

end) with v3 chemistry. Raw sequence data for 16S rRNA gene amplicon sequences are available at http://www.ebi.ac.uk/ under project number PRJEB30774.

**Cytokine measurement.** As described previously (20), cervicovaginal cytokine concentrations were measured using Luminex (Bio-Plex Pro Human Th17 cytokine panel, Bio-Rad Laboratories, Inc.), which included cytokines associated with Th17 cell differentiation (IL-1$\beta$, IL-6, TNF-$\alpha$, IL-23, and IL-33), those produced by Th17 cells (IL-17A, IL-17F, IL-21, and IL-22), and those associated with Th17 cell regulation (IL-25, IL-31, IFN-$\gamma$, and soluble CD40 ligand). Data were collected using a Bio-plex Suspension Array reader, as described previously (20).

**Flow cytometry.** Cervical cells were obtained from Digene cytobrushes, processed within 4 h, and stained immediately to measure T cell frequencies and activation by flow cytometry, as previously described (21). The gating strategy is shown in Fig. S1 in the supplemental material. The following panel of antibodies was included: APC-H7-CD3, BV605-CCR6, and APC-CCR5 (BD Biosciences); BV711-CD8, PE-CCR10, and Alexa-700-human leukocyte antigen (HLA) DR isotope (BioLegend); PE-Cy5.5-CD4 (Invitrogen); and PE-Cy7-cluster of differentiation 38 (CD38) (eBioscience, Inc.). LIVE-DEAD cell, Pacific-blue-CD14, and Pacific-blue-CD19 (Invitrogen) staining were performed to exclude dead cells, monocytes, and B-cells, respectively. Cells were acquired using an LSRFFortessa (BD Immunocytometry Systems). FlowJo v9.9.3 (FlowJo, LLC) was used for the data analysis.

**Data analysis.** All downstream analyses were performed in RStudio. Differences in baseline characteristics were tested using Pearson's chi-squared test or Fisher's exact test (when the expected value was <5) for count data, an unpaired Student's *t* test for differences in mean (parametric data), and an unpaired Mann–Whitney U test for differences in medians (nonparametric data) with *post hoc* testing. Linear multivariate regression models were used to adjust for covariates, including STIs, BV and HSV-2 serology, and randomization arm, and confounders were identified using a step-up model-building strategy. Incidence rates were calculated as the number of cases with *Candida* colonization or hypha formation at the 16-week follow-up visit per person per time in years (based on the time between the baseline and the 16-week follow-up visits) within each randomization arm. The differences in IR between arms were calculated by incidence-rate ratios (IRR). Downstream data analysis of previously generated 16S rRNA gene sequencing and cytokine data (20), including ecological diversity metrics and fold-change differences, was performed in R (v3.6.0) using the packages (57), NMF (58), and DESeq2 (59). The Data Integration Analysis for Biomarker discovery using Latent cOmponents (DIABLO) framework, as part of the mixOmics R Bioconductor package (30), was used for multiomics analyses integrating the cytokine, Th17 cell phenotype, and microbiota data sets, and to select features which accounted for the highest degree of variance between women with hypha formation.

**Data availability.** Raw sequence data for 16S rRNA gene amplicon sequences are available at the NCBI Sequence Read Archive under accession number PRJEB30774. All other data in this study are available from the corresponding authors upon reasonable request.

## SUPPLEMENTAL MATERIAL

Supplemental material is available online only.

**SUPPLEMENTAL FILE 1**, PDF file, 0.6 MB.

## ACKNOWLEDGMENTS

We thank the UChoose study team and all the young women who participated in the study. We also thank the Wellcome Centre for Infectious Diseases Research in Africa (CIDRI-Africa) for making their Applied Biosystems QuantStudio 7 Flex real-time PCR system available for our use. Computations were performed using facilities provided by the University of Cape Town's ICTS High Performance Computing team (http://hpc.uct.ac.za).

This study was supported by grants from the NIH (R01 HD083040), and Merck provided the NuvaRing.

H.B.J., A.-U.H., M.G., and S.D. conceived and designed the experiments. K.G. and L.-G.B. recruited the cohort. A.-U.H., M.G., I.K., C.B., and H.G. performed wet lab experiments. A.-U.H. and H.B.J. analyzed the data. A.-U.H. and H.B.J. wrote the manuscript. J.-A.S.P. and H.B.J. acquired funding. All authors reviewed and approved the final manuscript.

The authors declare no conflict of interests.

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
