## [Reviewer comments · Microbiology Spectrum]

Microbiology Spectrum

Persistent, asymptomatic colonization with *Candida* is associated with elevated frequencies of highly activated cervical Th17-like cells and related cytokines in the reproductive tract of South African adolescents

Anna Happel, Melanie Gasper, Christina Balle, Iyaloo Konstantinus, Hoyam Gamielidien, Smritee Dabee, Katherine Gill, Linda-Gail Bekker, Jo-Ann Passmore, and Heather Jaspan

Corresponding Author(s): Heather Jaspan, University of Washington

Review Timeline:

Submission Date:	September 20, 2021
Editorial Decision:	December 9, 2021
Revision Received:	January 11, 2022
Editorial Decision:	February 10, 2022
Revision Received:	February 15, 2022
Accepted:	February 25, 2022

Editor: Carolina Coelho

Reviewer(s): Disclosure of reviewer identity is with reference to reviewer comments included in decision letter(s). The following individuals involved in review of your submission have agreed to reveal their identity: Catherine Cai (Reviewer #1); peigen chen (Reviewer #2)

Transaction Report:

DOI: <https://doi.org/10.1128/spectrum.01626-21>

December 9, 2021

Dr. Anna Ursula Happel
University of Cape Town
Department of Pathology, Institute of Infectious Diseases and Molecular Medicine
Anzio Road, Observatory
Cape Town 7925
South Africa

Re: Spectrum01626-21 (Asymptomatic vaginal *Candida* colonisation is associated with elevated frequencies of highly activated cervical Th17 cells and related cytokines in the female genital tract of South African adolescents)

Dear Dr. Anna Ursula Happel:

I have now received two review reports on your work; their opinions is the article is of interest, but needs some modifications before publication.

Reviewer#1 notes that you need to justify phenotyping strategy to justify findings, and requests a reframing of paper to fit the data more closely. the critique of the limitations of this particular study (supports but is not conclusive evidence) needs to be very clear for readers, and only then future implications and how the data support a broader hypothesis can be discussed.

Reviewer #2 asks for some experiments and revisions and the majority of those can be done, please see below. Animal experiments are desirable but not required for resubmission.

Deposition of data in a repository is a requirement for resubmission.

Good luck, and thank you for submitting your manuscript to Microbiology Spectrum.

Link Not Available

Sincerely,

Carolina Coelho

Journals Department
Reviewer comments:

Reviewer #1 (Comments for the Author):

General

This was a fascinating study that assesses the relationship between oral contraceptive use, Candida colonization, the greater microbiome, and the immune profile of the female genital tract, all in the context of HIV acquisition risk. These are important questions that have direct clinical applications - e.g., do Candida infections need to be more aggressively treated in those at increased HIV risk. It would be very interesting to explore how symptomatic Candidiasis affects the vaginal immune environment (as no patients reported symptoms in this study), as well as what changes occur with anti-Candida therapies.

Major concerns

The authors use CCR6⁻,CCR10⁻ cells to indicate Th1/Th2-enriched cells, and CCR6⁺,CCR10⁻ cells to indicate Th17-enriched cells - is this commonly done? It is concerning that CCR6 can be expressed on Th1 cells as well, and no other makers (other than CD3/CD4) were used. The use of surface markers to designate differentiation is likely not as reliable as using transcription factors or cytokine expression, particularly when used in a relatively limited combination. Perhaps intracellular staining was not possible due to low starting cell number. Regardless, many conclusions are based on this approach, so authors should clarify why these markers were chosen and provide references supporting the use of these phenotyping methods.

The title of the paper asserts that colonization is associated with elevated frequencies of highly activated Th17 cells, but this does not seem to be supported by the data - doesn't Figure 2 show that at baseline, the frequency of Th17 cells overall is decreased in participants with colonization compared to those without? The data do show in figure 6 that percentages of highly activated Th17 cells are increased at 16 weeks in patients with colonization, but it's unclear what the significance of this is, as there is no comparison to those without colonization at 16 weeks.

Minor concerns and remaining questions

In line 163, it is proposed that lower frequencies of Th17-like cells in colonized patients is possibly due to AICD - this should be limited to the discussion section if the authors agree (already appears in lines 265-267), as it is possible but not supported by the data in the results section.

In line 260-261, authors comment that IL-17 and IL-22 are produced by MAIT and Th17 cells, but they are also produced by numerous other cell types - ILCs, NKTs, gamma delta T cells, etc., would be nice to have a mention of this.

In Figure 5A, is the change in hyphae prevalence with Net-En significant?

Technical corrections

Line 86 - missing the word "is" in "there clinical evidence that..."

Please ensure abbreviations for full terms are defined on first mention and vice versa. I believe "Net-En" in line 90 is not defined previously (full name appears later in line 92).

Line 210 - typo in the word "the"

Figure 1A may be mislabeled - the hyphae column is assigned to the STI legend, and the VVC column to the hyphae legend

Reviewer #2 (Comments for the Author):

This is an interesting study. In this study, the author described the relationship between asymptomatic vaginal Candida colonization and the frequency of cervical Th17 cells and related cytokines in the reproductive tract of adolescent females in South Africa. Finally, it is concluded that asymptomatic vaginal Candida colonization can increase Th1/Th17 cell frequency and related cytokines. The conclusion of the article has certain significance for reducing the risk of AIDS infection. However, I have the following questions.

- 1 . The number of samples in some groups seems to be too small (<30), which has a certain impact on the accuracy of microbial analysis.
- 2 . Many studies have confirmed that the different physiological stages of women have a great influence on microorganisms. In this study, are all microbial samples collected and unified.
- 3 . I think animal experiments are necessary to explain the relationship between asymptomatic Candida colonization and Th1/Th17 cell frequency.
- 4 . I did not find the address where the 16s sequencing raw data is placed

Staff Comments:

Preparing Revision Guidelines

Please return the manuscript within 60 days; if you cannot complete the modification within this time period, please contact me. If you do not wish to modify the manuscript and prefer to submit it to another journal, please notify me of your decision immediately so that the manuscript may be formally withdrawn from consideration by Microbiology Spectrum.

General

This was a fascinating study that assesses the relationship between oral contraceptive use, Candida colonization, the greater microbiome, and the immune profile of the female genital tract, all in the context of HIV acquisition risk. These are important questions that have direct clinical applications – e.g., do Candida infections need to be more aggressively treated in those at increased HIV risk. It would be very interesting to explore how symptomatic Candidiasis affects the vaginal immune environment (as no patients reported symptoms in this study), as well as what changes occur with anti-Candida therapies.

Major concerns

The authors use CCR6-,CCR10- cells to indicate Th1/Th2-enriched cells, and CCR6+,CCR10- cells to indicate Th17-enriched cells – is this commonly done? It is concerning that CCR6 can be expressed on Th1 cells as well, and no other makers (other than CD3/CD4) were used. The use of surface markers to designate differentiation is likely not as reliable as using transcription factors or cytokine expression, particularly when used in a relatively limited combination. Perhaps intracellular staining was not possible due to low starting cell number. Regardless, many conclusions are based on this approach, so authors should clarify why these markers were chosen and provide references supporting the use of these phenotyping methods.

The title of the paper asserts that colonization is associated with elevated frequencies of highly activated Th17 cells, but this does not seem to be supported by the data – doesn't Figure 2 show that at baseline, the frequency of Th17 cells overall is decreased in participants with colonization compared to those without? The data do show in figure 6 that percentages of highly activated Th17 cells are increased at 16 weeks in patients with colonization, but it's unclear what the significance of this is, as there is no comparison to those without colonization at 16 weeks.

Minor concerns and remaining questions

In line 163, it is proposed that lower frequencies of Th17-like cells in colonized patients is possibly due to AICD – this should be limited to the discussion section if the authors agree (already appears in lines 265-267), as it is possible but not supported by the data in the results section.

In line 260-261, authors comment that IL-17 and IL-22 are produced by MAIT and Th17 cells, but they are also produced by numerous other cell types – ILCs, NKTs, gamma delta T cells, etc., would be nice to have a mention of this.

In Figure 5A, is the change in hyphae prevalence with Net-En significant?

Technical corrections

Line 86 – missing the word “is” in “there clinical evidence that...”

Please ensure abbreviations for full terms are defined on first mention and vice versa. I believe “Net-En” in line 90 is not defined previously (full name appears later in line 92).

Line 210 – typo in the word “the”

Figure 1A may be mislabeled – the hyphae column is assigned to the STI legend, and the VVC column to the hyphae legend

Reviewer #1 (Comments for the Author):

General

This was a fascinating study that assesses the relationship between oral contraceptive use, Candida colonization, the greater microbiome, and the immune profile of the female genital tract, all in the context of HIV acquisition risk. These are important questions that have direct clinical applications - e.g., do Candida infections need to be more aggressively treated in those at increased HIV risk. It would be very interesting to explore how symptomatic Candidiasis affects the vaginal immune environment (as no patients reported symptoms in this study), as well as what changes occur with anti-Candida therapies.

Major concerns

The authors use CCR6-, CCR10- cells to indicate Th1/Th2-enriched cells, and CCR6+, CCR10- cells to indicate Th17-enriched cells - is this commonly done? It is concerning that CCR6 can be expressed on Th1 cells as well, and no other markers (other than CD3/CD4) were used. The use of surface markers to designate differentiation is likely not as reliable as using transcription factors or cytokine expression, particularly when used in a relatively limited combination. Perhaps intracellular staining was not possible due to low starting cell number. Regardless, many conclusions are based on this approach, so authors should clarify why these markers were chosen and provide references supporting the use of these phenotyping methods.

We acknowledge that for the conclusive definition of Th1, Th2 or Th17 cells a more complex flow cytometry panel with intracellular staining or evaluation of transcription factors would be required, which we have now mentioned in the discussion.

Unfortunately, the number of cervical cells obtainable from cytobrush samples is limited and thus intra-cellular staining after stimulation is not feasible. However, we would like to emphasize that we refer to these cell populations as Th17-enriched or Th17-like, indicating that these cell populations primarily contain Th17 cells but may contain a small proportion of other cell types. For further clarification, we are now referring to CCR6-CCR10- CD4+ T cells as such, rather than referring to this cell population as Th1/Th2-enriched cells. This panel has also previously been published in Clinical Infectious Diseases when evaluating the impact of hormonal contraceptives on cervical Th17 phenotype and function in the same cohort (Konstantinus et al. Impact of Hormonal Contraceptives on Cervical T-helper 17 Phenotype and Function in Adolescents: Results from a Randomized, Crossover Study Comparing Long-acting Injectable Norethisterone Oenanthate (NET-EN), Combined Oral Contraceptive Pills, and Combined Contraceptive Vaginal Rings, Clinical Infectious Diseases, Volume 71, Issue 7, 1 October 2020, Pages e76–e87, <https://doi.org/10.1093/cid/ciz1063>) and was based on similar panels previously published by others (McKinnon et al. Early HIV-1 infection is associated with reduced frequencies of cervical Th17 cells. J Acquir Immune Defic Syndr. 2015 Jan 1;68(1):6-12. doi: 10.1097/QAI.0000000000000389. PMID: 25296095. and Stieh et al. Th17 Cells Are Preferentially Infected Very Early after Vaginal Transmission of SIV in Macaques. Cell Host Microbe. 2016 Apr 13;19(4):529-40. doi: 10.1016/j.chom.2016.03.005. PMID: 27078070; PMCID: PMC4841252.). We have now cited these works to justify our approach.

The title of the paper asserts that colonization is associated with elevated frequencies of highly activated Th17 cells, but this does not seem to be supported by the data - doesn't Figure 2 show that at baseline, the frequency of Th17 cells overall is decreased in participants with colonization compared to those without? The data do show in figure 6 that percentages of highly activated Th17 cells are increased at 16 weeks in patients with colonization, but it's unclear what the significance of this is, as there is no comparison to those without colonization at 16 weeks.

We appreciate the reviewer for pointing this out. We have changed the title to clarify that persistent colonisation over 16 weeks was associated with elevated frequencies of highly activated cervical Th17 cells and related cytokines. To further support this statement, we have amended Figure 6 (and added Figure 7) to include participants who were not colonised over 16 weeks and those acquired or clearing *Candida*. We further have modified the result and discussion sections of the manuscript to interpret our data more cautiously.

Minor concerns and remaining questions

In line 163, it is proposed that lower frequencies of Th17-like cells in colonized patients is possibly due to AICD - this should be limited to the discussion section if the authors agree (already appears in lines 265-267), as it is possible but not supported by the data in the results section.

We have amended this section of the manuscript and limited these speculations to the discussion.

In line 260-261, authors comment that IL-17 and IL-22 are produced by MAIT and Th17 cells, but they are also produced by numerous other cell types - ILCs, NKTs, gamma delta T cells, etc., would be nice to have a mention of this.

We have edited this paragraph to reflect the cell types that contribute to IL-17 and IL-22 production more acutely.

In Figure 5A, is the change in hyphae prevalence with Net-En significant?

Despite a 3-fold increase in hyphae prevalence from baseline to week 16 in participants randomised Net-En, this was not significant ($p=0.116$). While this is stated in the result section, we have now added p-values to the figure and added a comment regarding this finding to the discussion.

Technical corrections

Line 86 - missing the word "is" in "there clinical evidence that..."

This has been corrected.

Please ensure abbreviations for full terms are defined on first mention and vice versa. I believe "Net-En" in line 90 is not defined previously (full name appears later in line 92).

We have reviewed the manuscript to avoid these errors.

Line 210 - typo in the word "the"

This has been corrected.

Figure 1A may be mislabeled - the hyphae column is assigned to the STI legend, and the VVC column to the hyphae legend

This has been corrected.

Reviewer #2 (Comments for the Author):

This is an interesting study. In this study, the author described the relationship between asymptomatic vaginal *Candida* colonization and the frequency of cervical Th17 cells and related cytokines in the reproductive tract of adolescent females in South Africa. Finally, it is concluded that asymptomatic vaginal *Candida* colonization can increase Th1/Th17 cell frequency and related cytokines. The conclusion of the article has certain significance for reducing the risk of AIDS infection. However, I have the following questions.

1. The number of samples in some groups seems to be too small (<30), which has a certain impact on the accuracy of microbial analysis.

We agree that the sample size for women with vulvo-vaginal candidiasis, or those acquiring or clearing *Candida*, is small and thus are limited with regards to the conclusions we can make. The sample size of the substudy was unfortunately limited to that of the parent study, and we have expanded on this limitation in the discussion.

2. Many studies have confirmed that the different physiological stages of women have a great influence on microorganisms. In this study, are all microbial samples collected and unified.

Studies have indeed shown that the stage of menstrual cycle may influence vaginal microbiota composition. Since all women were randomised to a certain hormonal contraceptive at baseline, participants were scheduled to return within 40 days of their screening visit for randomisation and samples collected during similar stages of their cycle (at baseline and 16 weeks after hormonal contraceptive initiation). This information has been added to the manuscript.

3. I think animal experiments are necessary to explain the relationship between asymptomatic *Candida* colonization and Th1/Th17 cell frequency.

We agree that animal experiments would be necessary to ultimately determine a causal relationship between *Candida* colonization and Th1/Th17 cell frequency. This is however beyond the scope of this manuscript. We have emphasized that our findings are based on associations and do not claim causality.

4. I did not find the address where the 16s sequencing raw data is placed

Our apologies, this has now been added to the manuscript.

February 10, 2022

Dr. Anna Ursula Happel
University of Cape Town
Department of Pathology, Institute of Infectious Diseases and Molecular Medicine
Anzio Road, Observatory
Cape Town 7925
South Africa

Re: Spectrum01626-21R1 (Persistent, asymptomatic colonisation with *Candida* is associated with elevated frequencies of highly activated cervical Th17-like cells and related cytokines in the reproductive tract of South African adolescents)

Dear Dr. Anna Ursula Happel:

You have addressed all reviewers' comments and included a clear section on limitations of the study, including sample size and microbial relationships.

Before full acceptance, please perform the edits requested by reviewer 1.

Thank you for submitting your manuscript to Microbiology Spectrum. As you will see your paper is very close to acceptance. Please modify the manuscript along the lines I have recommended. As these revisions are quite minor, I expect that you should be able to turn in the revised paper in less than 30 days, if not sooner. If your manuscript was reviewed, you will find the reviewers' comments below.

When submitting the revised version of your paper, please provide (1) point-by-point responses to the issues I raised in your cover letter, and (2) a PDF file that indicates the changes from the original submission (by highlighting or underlining the changes) as file type "Marked Up Manuscript - For Review Only". Please use this link to submit your revised manuscript. Detailed instructions on submitting your revised paper are below.

Link Not Available

Sincerely,

Carolina Coelho

Reviewer comments:

Reviewer #1 (Comments for the Author):

The reviewers have responded adequately to the previous round of suggestions and comments, and the revised version of the manuscript is clearer and better supported.
Only remaining minor suggestion is to edit line 34 of the abstract which currently reads "...significantly elevated concentrations of IL-22, IL-17A and IL-17F, all 34 produced by Th17 cells, but not of pro-inflammatory cytokines"... given that IL-17 can also be pro-inflammatory. Would consider revising more simply to "not of other cytokines such as IL-1 or IL-6."

Reviewer #2 (Comments for the Author):

For the problem of insufficient sample size, my concerns still cannot be solved. 16s rDNA data are characterized by high sparsity. My concern is whether the various relationships of microorganisms can be well discovered in the case of insufficient

sample size.

Preparing Revision Guidelines

- point-by-point responses to the issues I raised in your cover letter
- Upload a compare copy of the manuscript (without figures) as a "Marked-Up Manuscript" file.
- Each figure must be uploaded as a separate file, and any multipanel figures must be assembled into one file.
- Manuscript: A .DOC version of the revised manuscript
- Figures: Editable, high-resolution, individual figure files are required at revision, TIFF or EPS files are preferred

Please return the manuscript within 60 days; if you cannot complete the modification within this time period, please contact me. If you do not wish to modify the manuscript and prefer to submit it to another journal, please notify me of your decision immediately so that the manuscript may be formally withdrawn from consideration by Microbiology Spectrum.

Reviewer #1 (Comments for the Author):

The reviewers have responded adequately to the previous round of suggestions and comments, and the revised version of the manuscript is clearer and better supported. Only remaining minor suggestion is to edit line 34 of the abstract which currently reads "...significantly elevated concentrations of IL-22, IL-17A and IL-17F, all produced by Th17 cells, but not of pro-inflammatory cytokines"... given that IL-17 can also be pro-inflammatory. Would consider revising more simply to "not of other cytokines such as IL-1 or IL-6."

We have amended this section in the abstract as suggested. It now reads ..." hyphae presence was associated with significantly elevated concentrations of IL-22, IL-17A and IL-17F, all produced by Th17 cells, but not of other cytokines, such as IL-1 β or IL-6, after adjustment for confounders."

February 25, 2022

Dr. Heather B Jaspan
University of Washington
Department of Pediatrics and Global Health
Seattle, WA

Re: Spectrum01626-21R2 (Persistent, asymptomatic colonization with *Candida* is associated with elevated frequencies of highly activated cervical Th17-like cells and related cytokines in the reproductive tract of South African adolescents)

Dear Dr. Heather B Jaspan:

Thank you for submitting this article to Spectrum.

Your manuscript has been accepted, and I am forwarding it to the ASM Journals Department for publication. You will be notified when your proofs are ready to be viewed.

Please update the methods to have a "Data availability" paragraph, as per journal style.

Sincerely,

Carolina Coelho
Editor, Microbiology Spectrum

Journals Department
Supplemental Material for Publication: Accept